# miR-625-3p and lncRNA GAS5 in Liquid Biopsies for Predicting the Outcome of Malignant Pleural Mesothelioma Patients Treated with Neo-Adjuvant Chemotherapy and Surgery

**DOI:** 10.3390/ncrna5020041

**Published:** 2019-06-17

**Authors:** Jelena Kresoja-Rakic, Adam Szpechcinski, Michaela B. Kirschner, Manuel Ronner, Brenda Minatel, Victor D. Martinez, Wan L. Lam, Walter Weder, Rolf Stahel, Martin Früh, Ferdinando Cerciello, Emanuela Felley-Bosco

**Affiliations:** 1Laboratory of Molecular Oncology, Department of Thoracic Surgery, University Hospital Zürich, 8091 Zürich, Switzerland; j.rakic@access.uzh.ch (J.K.-R.); a.szpechcinski@igichp.edu.pl (A.S.); manuel.ronner@usz.ch (M.R.); 2Department of Thoracic Surgery, University Hospital Zürich, 8091 Zürich, Switzerland; Michaela.Kirschner@usz.ch (M.B.K.); walter.weder@usz.ch (W.W.); 3BC Cancer Research Centre, Vancouver, BC V5Z 1L3, Canada; bminatel@bccrc.ca (B.M.); Victor.Martinez@iwk.nshealth.ca (V.D.M.); WanLam@bccrc.ca (W.L.L.); 4IWK Health Centre, Halifax, NS B3K 6R8, Canada; 5Comprehensive Cancer Center Zürich, University Hospital Zürich, 8091 Zürich, Switzerland; rolf.stahel@usz.ch; 6Cantonal Hospital of St. Gallen, 9007 St. Gallen, Switzerland; martin.frueh@kssg.ch; 7Department of Medical Oncology/Hematology, University of Bern, CH-3000 Bern, Switzerland; 8Clinic of Oncology, University Hospital Zürich, 8091 Zürich, Switzerland

**Keywords:** circulating noncoding RNA, miR-625-3p isomiRs, *GAS5*, malignant pleural mesothelioma, liquid biopsy

## Abstract

Combining neo-adjuvant chemotherapy and surgery is part of multimodality treatment of malignant pleural mesothelioma (MPM), but not all patients benefit from this approach. In this exploratory analysis, we investigated the prognostic value of circulating miR-625-3p and lncRNA GAS5 after neo-adjuvant chemotherapy. 36 MPM patients from the SAKK 17/04 trial (NCT00334594), whose blood was available before and after chemotherapy were investigated. RNA was isolated from plasma and reverse transcribed into cDNA. miR-16-5p and β-actin were used as a reference gene for miR-625-3p and GAS5, respectively. After exclusion of samples due to hemolysis or RNA degradation, paired plasma samples from 32 patients before and after chemotherapy were further analyzed. Quantification of miR-625-3p levels in all 64 samples revealed a bimodal distribution and cloning and sequencing of miR-625-3p qPCR product revealed the presence of miR-625-3p isomiRs. Relative change of the circulating miR-625-3p and GAS5 levels after chemotherapy showed that increased circulating miR-625-3p and decreased GAS5 was significantly associated with disease progression (Fisher’s test, *p* = 0.0393). In addition, decreased levels of circulating GAS5 were significantly associated with shorter overall and progression-free survival. Our exploratory analysis revealed a potential value of circulating non-coding RNA for selection of patients likely to benefit from surgery after platinum-based adjuvant chemotherapy.

## 1. Introduction

Malignant pleural mesothelioma (MPM) is an aggressive cancer of the pleura for which effective treatments are limited. Platinum-based induction chemotherapy followed by surgery can achieve prolonged survival [1,2,3]. The accurate selection of patients who may profit from surgery is crucial considering the aggressive and mostly fatal nature of the cancer and the morbidity (and mortality), which may be associated with the surgical intervention. Efforts are ongoing to define objective and unbiased prognostic scores, which may help for the selection of candidate patients for surgery [4,5]. These algorithms are mainly based on clinical characteristics and the biology of the tumor is only marginally taken into consideration, if at all. In this sense, tumor biomarkers would be important because they would permit to integrate information on the biology of the tumor within clinical decisions algorithms. Ideally, they would allow the recognition of cancer subtypes refractory to treatment for which aggressive procedures like surgery would be less effective. Biomarkers from the blood would be best suited for this purpose because they can be accessed easily and repetitively. In recent years non-coding RNA (ncRNA) have been identified as a novel opportunity for MPM biomarkers with prognostic as well as predictive potential [6,7,8,9,10]. Further, ncRNA can be detected from the blood [11], where they can serve as circulating biomarkers for MPM (reviewed in [12]). In our work we intended to perform an exploratory study to investigate the potential of ncRNA from the blood to help identify patients who will profit from surgery after platinum-based chemotherapy. To do so we focused on miR-625-3p, which we had been previously describing as a circulating biomarker in mesothelioma [13] and GAS5, a long non-coding RNA (lncRNA) that we had functionally characterized in MPM [14]. Additionally, high levels of miR-635-3p are associated with poor response to oxaliplatin in colon cancer [15] and expression of miR-625-3p induces oxaliplatin resistance in colon cancer cells [16] and promotes the proliferation of thyroid cancer cells [17]. The *GAS5* gene is a so-called host gene for small nucleolar RNA (snoRNA) and it is encoded at locus 1q25. It has up to 12 exons and 10 box C/D snoRNAs within its alternative introns together with conserved 5′-terminal oligopyrimidine tract (5′ TOP) [18]. GAS5 is named based on the finding that its expression levels increased upon cell growth arrest induced after serum starvation [19] or as the result of rapamycin-induced cell cycle arrest [18]. In this study, we have investigated changes in the blood level of miR-625-3p and GAS5 before and after platinum chemotherapy in a unique cohort of MPM patients treated with neoadjuvant chemotherapy followed by extrapleural pneumonectomy (EPP) and randomized for radiotherapy [20]. Next, we have correlated the blood level changes of the two ncRNAs with patients’ survival. Our work provides an exploratory observation on the prognostic role of miR-625-3p and GAS5 after adjuvant chemotherapy, which may represent an additional help for the selection of candidate patients for surgery

## 2. Results

### 2.1. Plasma Levels of Circulating miR-625-3p and GAS5 in the Cohort of MPM Patients

Circulating miR-625-3p and GAS5 ncRNAs were measured in 36 MPM patients for which plasma samples were available before and after chemotherapy. The characteristics of patients are summarized in Table 1. Briefly, 29 patients had epithelioid, 2 had the sarcomatoid, 4 mixed, and 1 had spindle cell shaped mesothelioma. 33 were male and 3 were female patients.

To normalize miRNA and lncRNA measurements, miR-16-5p [21] and β-actin were used as the plasma-based reference genes, respectively (Appendix A). Two samples were excluded due to RNA degradation. The coefficient of variation (CV) of Ct values across the remaining samples was 4.9% for miR-16-5p and 5.7% for β-actin. The low variation in reference genes demonstrated the stability of the plasma samples and an equal input amount of plasma-isolated RNA.

We next performed a quality control for cellular contamination, as hemolysis can occur due to blood processing and affects the miRNA measurement [21]. The levels of the hemolysis specific miRNA miR-451a and miR-23a-3p were measured. In two samples, Delta Ct difference between miR-23a-3p and miR-451 (miR-23a–miR-451-3p) was more than five due to hemolysis contamination [22] and we therefore excluded these samples.

We next assessed the relative levels of circulating miR-625-3p in the remaining 32 pairs of plasma samples (Figure 1a). MiR-625-3p levels displayed bimodal distribution with 16 patients presenting relative levels (ΔCt) between 4.61 and 9.28 before and after chemotherapy (miR-625-3p^low^) and 16 patients with relative levels between −0.07 and −3.95 (miR-625-3p^high^) (Figure 1a).

To exclude the possibility that the observed bimodal distribution of circulating miR-625-3p was due to the cross reactivity and detection of other miRNA species, q-PCR products from 2 samples, one which belonged to the miR-625-3p^high^ (below median of ∆Ct) and another which belonged to the miR-625-3p^low^ (above median of ∆Ct) group were cloned and sequenced. We found only miR-625-3p along with isoforms, called isomiRs, which have mostly variations around their 3′ end [23] (Appendix A), demonstrating that the whole population of miR-625-3p accounts for the observed bimodal distribution. Interestingly, isomiRs only were detected in the clones derived from the sample in the miR-625-3p^low^ group, indicating the possibility that in some patients circulating miR-625-3p isomiRs may be more abundant compared to canonical archetype sequence.

Analysis [24] of an independent cohort of small RNA-seq data derived from mesothelioma samples from untreated patients generated by The Cancer Genome Atlas (TCGA) not only confirmed the presence of these miR-625-3p isomiRs detected in circulation (Figure 1b) as well as revealed that altogether miR-625-3p isomiRs counts are abundant in mesothelioma tissue.

GAS5 levels showed continuous distribution among samples with decreased abundance after chemotherapy in 9 patients and increase in 22 patients (Figure 1c), while no change was observed in one patient. Taken together, our results provide evidence that platinum-based chemotherapy can induce changes in the plasma level of circulating miR-625-3p and GAS5 and that these changes are not uniform upon treatment but rather specific for each patient.

### 2.2. Post-Chemotherapy Changes in the Plasma Level of miR-625-3p and GAS5 and Response to Treatment

Next, we were interested in investigating if the changes observed in the plasma levels of circulating miR-625-3p or GAS5 after platinum-based chemotherapy may associate with treatment response. The cohort included 24 patients with partial response (PR) or stable disease (SD) after platinum based neo-adjuvant chemotherapy and 8 patients with progressive disease (PD). Chemotherapy induced changes in plasma levels of miR-625-3p were not significantly different between the two groups of PD and PR/SD (*p* = 0.6851) and showed an average fold increase of 2.7 and decrease of 2.0 (Figure 2a). After treatment, the levels of miR-625-3p were increased in the plasma of 11 patients with PR/SD (45.8% of the PR/SD subgroup) and in 5 patients with PD (62.5% of the PD subgroup) and decreased in 13 patients with PR/SD (54.2% of the subgroup) and 3 patients with PD (37.5% of the subgroup).

Changes of GAS5 levels in plasma were not significantly different (*p* = 0.1845) between the group of PD and PR/SD (Figure 2b). The average fold change for the increase of GAS5 was 21.5 and for the decrease it was 4.2. 50% of the patients with PD had an increase and 50% had a decrease of GAS5 plasma levels after chemotherapy (4 patients showed an increase and 4 patients showed a decrease of GAS5 after treatment). In the subgroup of patients with PR/SD after treatment, we observed an increase in GAS5 in 18 patients after treatment (75% of the subgroup) and a decrease or no change in 6 patients (25% of the subgroup). Considering changes in miR-625-3p and GAS5 together, we could observe that the combination of miR-625-3p increase and GAS5 decrease after chemotherapy was significantly associated with disease progression (Fisher’s test, *p* = 0.0393) (Figure 2c). These observations led to the hypothesis that plasma levels of miR-625-3p and GAS5 may associate with particular biological subtypes of MPM with different sensitivity to chemotherapy and aggressiveness.

### 2.3. Decrease in Circulating GAS5 Associates with Shorter Progression-Free and Overall Survival

Finally, we were interested in investigating if variations of circulating miR-625-3p and GAS5 after chemotherapy may have a prognostic value as suggested from our observations above. We therefore investigated survival in the subgroups of patient with miR-625-3p low or high and in the subgroups of patients with GAS5 low or high. For the subgroups of miR-625-3p we could not observe any significant differences in overall or progression free survival between patients with low versus high miR-625-3p levels. Focusing on GAS5 instead, we observed that patients with decreased levels of GAS5 after chemotherapy had a median overall survival of 7.8 months which was significantly shorter than median overall survival of patients with GAS5 increase (11.8 months, *p* = 0.0308, Figure 3a). Also, progression free survival was significantly shorter for patients with decreased GAS5 after chemotherapy (*p* = 0.0096, Figure 3b). These results provided additional evidence for the prognostic role in particular of GAS5 in MPM after chemotherapy treatment.

## 3. Discussion

In this study, we explore miR-625-3p and GAS5 as circulating biomarkers for predicting treatment outcome for MPM patients undergoing neo-adjuvant chemotherapy and surgery. We observed that increased levels of miR-625-3p together with decreased levels of GAS5 after platinum doublets were more frequent in the plasma of patients who were refractory to chemotherapy. Our results in blood reflected similar recent observations from the TCGA consortium in MPM tissue where low levels of miR-625-3p and high levels of GAS5 were associated with a better clinical outcome [25]. In the TCGA dataset miR-625-3p expression positively correlated with a signature of epithelial-mesenchymal transition (R = 0.32, *p* = 0.0049, FDR0.0232) [25]. Epithelial to mesenchymal transition could contribute at least in part, to the chemo-resistance observed in MPM patients with increased levels of miR-625-3p after chemotherapy. GAS5 is involved in growth control and apoptosis and its expression is predominantly identified in quiescent MPM tumor cells [14,26]. Reduced levels of GAS5 transcripts are reported to be indicative of poor prognosis in several types of cancer [26,27,28,29]. In accord, in our study overall survival and progression free intervals were significantly shorter for patients with decreased plasma levels of GAS5 after neoadjuvant chemotherapy. Based on our observation, both miR-625-3p and GAS5 may represent interesting blood-based biomarkers for clinical decisions. The associations of the two ncRNAs with response to treatment after chemotherapy could help in the differentiation of responders from non-responders, and if integrated within the radiological monitoring of MPM patients, it may aid to the selection of the best suited surgery approaches. Also, considering that changes in levels of GAS5 may predict outcome after chemotherapy and surgery, it may hold potential in guiding treatment decisions. If integrated within the current clinical decisions tree, it may help to identify in particular those patients with an expected poor outcome after therapy and for whom aggressive treatments will be less beneficial. We have to acknowledge here that our observations are based only on a limited number of patients. The size of our cohort was dictated by the discovery and hypothesis generating intent of our investigation. We believe that further validation in an independent next clinical study on larger number of samples would be highly relevant for the clinical practice of MPM. Ideally, we would extend our cohort to include patients treated with EPP but also pleural decortication, in order to assess the biomarkers predicted outcome for different surgical strategies.

Intriguingly, we observed a bimodal distribution for miR-625-3p. Molecular profiles with bimodal gene expression have consequences for biomarker discovery since they allow setting up a cutoff and have been used in tissues and circulating tumor cells [30,31,32]. Bimodal distributions of other circulating miRs have already been observed in other pathological conditions, such as myelodisplatic syndrome [33]. However, it had not been observed in a previous study in mesothelioma where we used stem-loop methodology to detect miR-625-3p [13]. Therefore, the bimodal distribution observed is possibly linked to the method of detection applied in this study, which is able to detect a larger number of isomiRs [34]. IsomiRs profiles are able to categorize individuals based on characteristics, such as sex, race, and population [35] and can distinguish amongst tissue types and disease subtypes [36,37]. It has been estimated that only 8% of the miRNA isoforms present in serum are the canonical miRNA form [38]. IsomiRs with shifts on their 5′ end alters the miRNA seed sequence, resulting in changes in target profiles that can target largely non-overlapping groups of mRNAs [39], however, while the existence of 5′ variant of miR-625-3p has already been described in MPM tissue [40], we found circulating 3′ variants in the few clones sequenced. 3′ end modifications modulate miRNA processing, stability, argonaute loading and targeting effectiveness [41,42,43,44]. The isomiRs that we identified are also present in MPM tissue and we describe that miR-625-3p isomiRs are altogether more abundant than their canonical counterpart. Future studies focused on a more in depth analysis of circulating miR-625 isomiRs will be required to better understand their functions, which are not yet fully understood.

In summary, this exploratory study demonstrates the feasibility of investigating circulating non-coding RNA in liquid biopsies. We observed that changes in levels of circulating miR-625-3p and GAS5 associated with response to platinum-based chemotherapy and predicted outcome after neo-adjuvant treatment and surgery in MPM. These novel blood-based biomarkers may be helpful for clinical treatment decisions in MPM.

## 4. Materials and Methods

### 4.1. Blood Collection and Preparation of Plasma

All subjects gave their informed consent for inclusion before they participated in the study. The study was conducted in accordance with the Declaration of Helsinki, and the protocol was approved by the Ethics Committee of the Zurich University Hospital (KEK-StV-Nr. 24/05). Venous blood was collected in 10 mL Vacutainer Plus tubes with K2EDTA additive (BD Biosciences) from 36 mesothelioma patients that were enrolled in the trial SAKK 17/04 (NCT00334594) [20]. Blood samples from MPM patients were collected before and after chemotherapy treatment. Within 1 h of collection, the blood was centrifuged in the original vials for 10 min at 3000 rpm (approx. 1600 x g) at room temperature. Plasma was carefully transferred to a new 15 mL non-pyrogenic Falcon tube and centrifuged again for 10 min at 3000 rpm at room temperature in order to pellet any residual cells. Cell-free plasma was then aliquoted to 0.5 mL volumes in 1.5 mL RNase-free Eppendorf tubes and stored at −80 °C for further processing.

### 4.2. RNA Extraction from Plasma

Total RNA enriched with circulating miRNA was extracted from 200 μL plasma using the miRNeasy Mini kit (Qiagen, Hilden, Germany) following the manufacturer’s instruction. Plasma samples were thawed completely on ice and mixed gently by pipetting immediately prior to aliquoting for RNA isolation and to evenly disperse any particulates present. 5 volumes of QIAzol Lysis Reagent was added followed by vortexing for dissociation of nucleoprotein complexes and 5 min incubation at room temperature. One μg MS2 carrier RNA (Roche, Basel, Switzerland) was added before addition of 1 volume of chloroform, followed by vigorously vortexing and centrifugation for 15 min at 12,000× *g* at 4 °C for subsequent phase separation. The upper aqueous phase containing total RNA was carefully transferred to a new collection tube, and 1.5 volume of 100% ethanol was added. The sample was passed through the RNeasy Mini spin column and purified by several washing steps. In order to obtain a maximum yield, the RNA was eluted from the spin column using two subsequent RNase-free water aliquots (30 μL each), and stored at −80 °C for further processing. To control for extraction efficacy, synthetic, non-human *C. elegans* miR-39-3p mimic (Qiagen) was introduced in amount of 2.8 × 10^8^ copies to plasma samples after the addition of the QIAzol Lysis Reagent. This mimic went through the entire RNA isolation process and was ultimately measured by RT-qPCR in the final RNA eluate, providing an internal control of technical variations between samples. The coefficient of variation was 2.7%.

### 4.3. Reverse Transcription of miRNA and qRT-PCR Measurement

The reverse transcription of miRNAs isolated from plasma was performed using miScript II RT Kit (Qiagen) using 5 μL of total RNA. In the Qiagen’s miScript system, all mature miRNAs are polyadenylated by poly(A) polymerase and reverse transcribed into cDNA library using oligo-dT primers. The oligo-dT primers have a 3′ degenerate anchor and a universal tag sequence on the 5’ end, allowing amplification and detection of multiple mature miRNAs in the real-time PCR step. In our study the cDNA synthesis was performed using the miScript HiSpecBuffer (for mature miRNA detection only) and 5 μL of the RNA isolated from plasma was used in a total reaction volume of 20 μL at the following temperature conditions: 60 min at 37 °C followed by 5 min in 95 °C. After reverse transcription, the cDNA was diluted into 1:20 ratio. Obtained cDNA was used immediately in qPCR reactions or stored at −20 °C for further use within 1 week. To perform qPCR, 3 μL was used from diluted cDNA in a total reaction volume of 10 μL which was set up in triplicates. Amplification was performed using specific miScript Primer Assays specific for each miRNA (Appendix A) together with miScript SYBR GreenPCR Kit (Qiagen) according to manufacturer’s instruction. Since hemolysis can happen during plasma preparation and affect the quantification of circulating miRNA, we measured plasma levels of miR-451-3p and miR-23a as hemolysis indicators [22]. Samples which had ΔCt (miR-23a – miR-451-3p) more than 5 Ct cycles were excluded as samples contamination induced by hemolysis.

5 μL of total RNA was taken to prepare cDNA using Qiagen QuantiTect^®^ Reverse Transcription. cDNA was amplified using Power SYBR Green PCR Master Mix (Life technologies 4367659) and gene specific primers for GAS5 [14] and β-actin 5′ GGACCTGACTGACTACCTCAT and 5′-CGTAGCACAGCTTCTCCTTAAT-3′. β-actin was selected as normalizer for circulating lncRNA because it has been used in other cancer related studies [45,46] and seems relatively stable compared to non-uniform expression pattern of Sno [38] or rRNA [47]. Relative mRNA levels (ΔCt values) were determined by comparing the PCR cycle thresholds (Ct) between GAS5 and reference gene β-actin. The relative change in the circulating levels of miR-625-3p or GAS5 (ΔΔCt) was calculated as following: ΔCt _after chemotherapy_ − ΔCt _before chemotherapy_. Based on the relative change, the patient cohort was stratified as miR-625-3p^increase^ or miR-625-3p^decrease^, and GAS5^increase^ or GAS5^decrease^. Increase was defined as a relative change ΔΔCt < 0 whereas decrease was defined for those that have ΔΔCt > 0.

For all q-PCR “No-template” samples were included as negative controls. The qPCR reactions were set up manually and run on the 7500 Fast Real-Time PCR System thermal cycler (Applied Biosystems, ThermoFisher Scientific, Reinach, Switzerland) and determined using the 7500 Software 2.0.4 (Applied Biosystems). The specificity of all the q-PCR products was verified by dissociation curves and by running on 4% agarose gel.

### 4.4. TA Cloning

For cloning and sequencing of q-PCR obtained miR-625-3p products, TA cloning technique was performed using pCR4-TOPO vector (Invitrogen, 450030, ThermoFisher Scientific, Reinach, Switzerland). To set up a TOPO-TA cloning reaction, the following reagents were mixed: Fresh 1 μL q-PCR product, 1 μL salt solution (1.2 M NaCl, 0.06 M MgCl_2_), 1 μL pCR4-TOPO vector, 3 μL water, and incubated 30 min at room temperature. For positive selection, ccdB screen approach was used. Briefly, bacteria NEB10-β (New England BioLabs C3019l, Ipswich, MA, USA) were transformed with 2 μL of TOPO-TA cloning reaction and plated on LB-ampicillin plates after which single colonies were picked and DNA plasmids were extracted. Isolated DNA plasmids were sequenced using either T7 or M13 primers.

### 4.5. miR-625 isomiR Detection in Mesothelioma Samples from TCGA

Small RNA sequencing data from a cohort of 87 MPM samples processed by The Cancer Genome Atlas (TCGA) were obtained through the Genomic Data Commons (GDC) (https://portal.gdc.cancer.gov/legacy-archive/search/f, dbgap Project ID: 6208). All small RNA sequencing data that had been generated using the Illumina HiSeq2000 platform was processed according to a previously described pipeline [48,49]. Briefly, aligned sequencing reads (.bam files) retrieved from GDC were converted to unaligned reads (.fastq files) and trimmed based on Phred quality scores (≥20). Trimmed unaligned reads were then submitted to the online platform miRMaster (https://ccb-compute.cs.uni-saarland.de/mirmaster) [50]. This platform aligns the reads to the newest version of the human genome (hg38) using the Bowtie aligner and quantifies miRNA precursors annotated in miRBase v.21, while allowing for one nucleotide mismatch (http://www.mirbase.org/) [51]. The isomiRs detection performed by miRMaster relies on the mapping of sequencing reads to annotated miRNA precursors while allowing for up to two 5′/3′ additions and one mismatch in between. Up to two nucleotide variations at the 5′ end and up to five at the 3′ end are also allowed. Finally, all read counts were normalized to reads per million (RPM).

### 4.6. Statistical Analysis

All statistical analyses were performed using the GraphPad Prism 5 software (GraphPad Software, Inc, San Diego, CA, USA). Error bars represent standard deviation of the mean. Correlation degree between basal levels of circulating miR-625-3p and *GAS5* levels was determined by Pearson’s correlation analysis. Statistical analysis of Kaplan–Meier curve was performed using Log-rank test. The difference between two response groups PD (progressive disease) and PR/SD (partial response/stable disease) was analyzed by Fisher’s exact test (two-sided). All p that were < 0.05 were considered as statistically significant.

## Figures and Tables

**Figure 1 ncrna-05-00041-f001:**
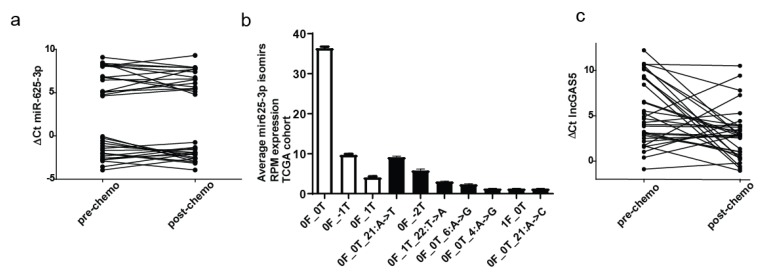
Relative circulating levels of miR-625-3p and GAS5 in plasma samples from 32 malignant pleural mesothelioma (MPM) patients. (**a**) Relative circulating levels (ΔCt) of mir-625-3p were determined by q-PCR in blood collected before and after receiving chemotherapy. Note that higher Ct value indicates lower expression levels. Two groups were defined: miR-625-3p^high^ (below median of ∆Ct) group and miR-625-3p^low^ group (above median of ∆Ct). (**b**). Average expression levels of the major miR-625-3p isomiRs in tumor samples derived from the TCGA MPM cohort. White bars indicate the tissue levels of the isomiRs, which we also detected in our plasma samples. The nomenclature of isomiRs indicates the position and number of nucleotides alterations with respect to the archetypes’ sequence. The first number indicates the relative position of the isomiR’s 5′ terminus (F) with respect to the archetype’s 5′ end; whereas the second number indicates the analogous relationship for the isomiR’s and the archetypes’s 3′ termini (T). A positive sign (+) indicates that the isomiR’s terminus is downstream from the archetype terminus (in 5′→3′ direction), while a negative sign (−) indicates that the isomiR’s terminus is upstream of the archetype’s terminus. Using this notation, miR-625-3p 0F_0T denotes the archetype miRNA. (**c**) Relative circulating levels (ΔCt) of GAS5 in blood collected before and after receiving chemotherapy shows a continuous distribution.

**Figure 2 ncrna-05-00041-f002:**
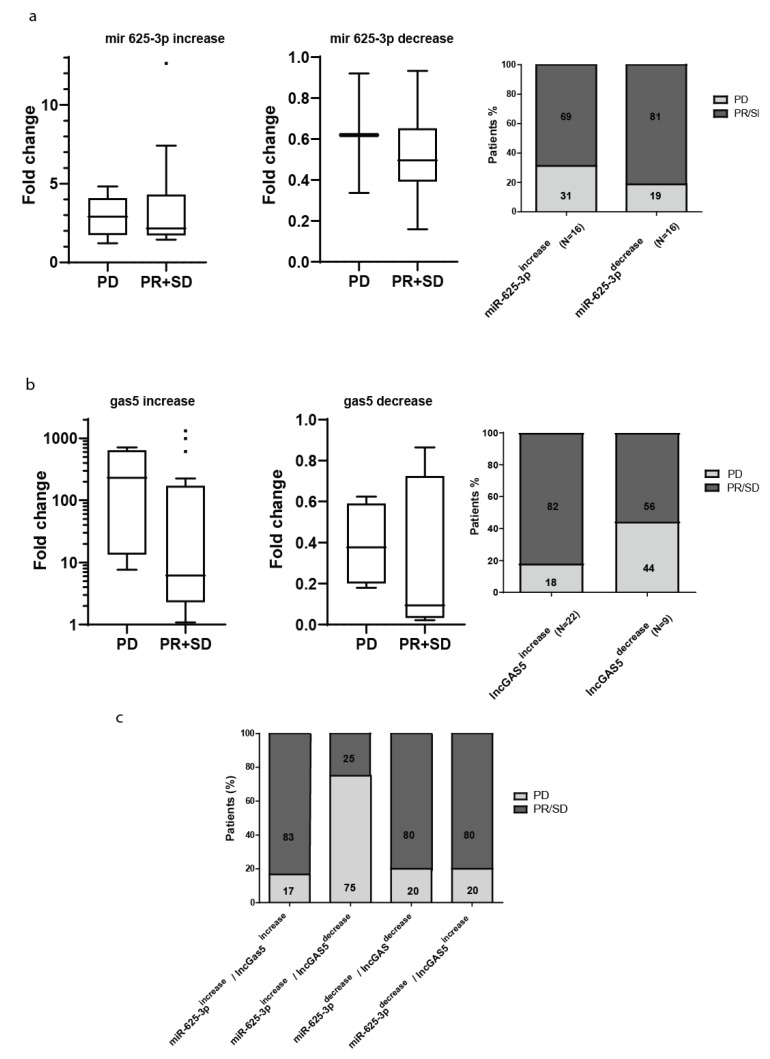
Increase of circulating miR-625-3p in combination with decrease of circulating GAS5 associates with disease progression. Relative changes of circulating miR-625-3p (**a**) and GAS5 (**b**) before and after chemotherapy defined patients groups where these circulating nucleic acids displayed increased or decreased levels. Neither relative changes in miR-625-3p nor in GAS5, was significantly (Fisher’s test) associated with response to chemotherapy defined as partial response (PR) or stable disease (SD) and compared to progressive disease (PD). The Tukey box plots of fold change expression have median values represented by the line within the boxes and the lower and upper limit of the boxes denote the 25th and 75th percentile while dots represent outliers (**c**). Combined increasing circulating levels of miR-625-3p and decreasing circulating levels of GAS5 were significantly (*p* = 0.0393, Fisher’s test) associated with the progressive disease (PD) after chemotherapy.

**Figure 3 ncrna-05-00041-f003:**
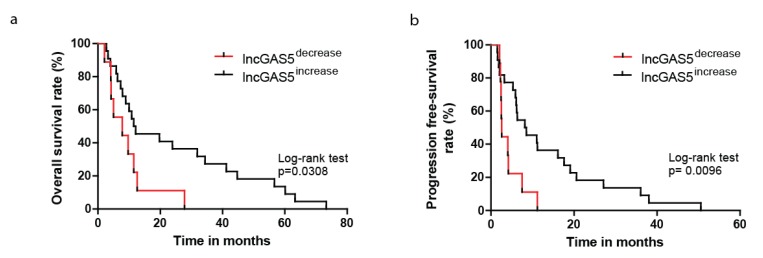
Decrease of circulating levels of GAS5 is associated with the worse patient outcome. Patients with decreased circulating GAS5 after chemotherapy had a shorter overall survival (**a**) and a shorter progression-free survival (**b**).

**Table 1 ncrna-05-00041-t001:** Characteristics of mesothelioma patients.

Variable	Total	Epithelioid	Sarcomatoid	Mixed	Other
Number	36	29	2	4	1
Age (Mean ± SD)	62.4 ± 5.04	62.5 ± 4.4	61.5 ± 0.7	62.6 ± 2.06	60
Gender (M/F)	33/3	26/3	2/0	4/0	1/0

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
