# Peer review of "miR-625-3p and lncRNA GAS5 in Liquid Biopsies for Predicting the Outcome of Malignant Pleural Mesothelioma Patients Treated with Neo-Adjuvant Chemotherapy and Surgery"

_ncrna, 2019, doi:10.3390/ncrna5020041_

Round 1

Reviewer 1 Report

The manuscript by Kresoja-Rakic and coworkers describes the use of a binomial approach combining a miRNA and a lncRNA as biomarkers for the outcome of malignant pleural mesothelioma in the context of neo-adjuvant therapy. The work is technically sound and within the scope of the journal. I sincerely believe that it will be interesting for the journal readers.

I acknowledge the technical care showed by the authors in checking the quality of the obtained RNA samples by considering all the contaminating factors including hemolysis.

Specific comments:

1.- The authors should comment about the limitations of the study, taking into account the small number of samples used.

2.- The normalization of GAS5 lncRNA was made taking into account the levels of beta-actin transcript. Please discuss what are the reasons for the selection of this normalizer, and the advantages/disadvantages in comparison with other circulating ncRNAs such as snoRNAs or rRNAs.

3.- In the discussion the authors described the presence of a "bimodal distribution" of miR-625-3p levels. Please clarify the meaning of "bimodal" in the context of circulating biomarkers and further discuss the consequences of this distribution. I sincerely think that considering that this bimodal distribution could be a consequence of the detection method, is a little bit speculative. Please give references in other experimental context that could support this idea.

4.- The authors discuss about the possible role of the interesting findings related with the circulating isoforms of miR-625-3p, however it is not clear if the authors considered the levels of these isoforms for the description of the proposed biomarkers. Are these isomiRs a direct consequence of the dysregulated expression pattern of cancer cells, specifically in the context of malignant mesothelioma? Is this phenomenon observed in other circulating miRNAs in other cancer types?.

Author Response

Reviewer  #1

We thank Reviewer #1 for having highlighted that The work is technically sound and within the scope of the journal. I sincerely believe that it will be interesting for the journal readers. I acknowledge the technical care showed by the authors in checking the quality of the obtained RNA samples by considering all the contaminating factors including hemolysis”. Additionally, we appreciate the valuable comments and suggestions provided by the Reviewer, which are addressed below.

1.- The authors should comment about the limitations of the study, taking into account the small number of samples used.

In order to comment about the limitations of our exploratory study we have added the following sentences (line 214):

“We have to acknowledge here that our observations are based only on a limited number of patients. The size of our cohort was dictated by the discovery and hypothesis generating intent of our investigation. We believe that further validation in an independent next clinical study on larger number of samples would be highly relevant for the clinical practice of MPM. Ideally we would extend our cohort to include patients treated with EPP but also pleural decortication, in order to assess the biomarkers predicted outcome for different surgical strategies.“

2.- The normalization of GAS5 lncRNA was made taking into account the levels of beta-actin transcript. Please discuss what are the reasons for the selection of this normalizer, and the advantages/disadvantages in comparison with other circulating ncRNAs such as snoRNAs or rRNAs.

The following sentence has been added line 297:

“β-actin was selected as normalizer for circulating lncRNA because it has been used in other cancer related studies [41,42] and seems relatively stable compared to non uniform expression pattern of sno [34] or rRNA [43]”

3.- In the discussion the authors described the presence of a "bimodal distribution" of miR-625-3p levels. Please clarify the meaning of "bimodal" in the context of circulating biomarkers and further discuss the consequences of this distribution. I sincerely think that considering that this bimodal distribution could be a consequence of the detection method, is a little bit speculative. Please give references in other experimental context that could support this idea.

The reviewer is right in pointing out that bimodal gene expression is useful because it allows classification of samples/patients according to a distinct expression state. We have therefore added the following sentence to line 220:

“Molecular profiles with bimodal gene expression have consequences for biomarker discovery since they allow setting up a cutoff and have been used in tissues and circulating tumor cells [29-31]. “

We put forward the hypothesis that the bimodal distribution could be a consequence of the detection method because using the stem loop method we do not observe such distribution.

4.- The authors discuss about the possible role of the interesting findings related with the circulating isoforms of miR-625-3p, however it is not clear if the authors considered the levels of these isoforms for the description of the proposed biomarkers. Are these isomiRs a direct consequence of the dysregulated expression pattern of cancer cells, specifically in the context of malignant mesothelioma? Is this phenomenon observed in other circulating miRNAs in other cancer types?.

Several recent publications have demonstrated that stem loop methodology for miR detection does not detect isomiRs. By using a different method, we detected archetype and isomiRs sequences so we cannot for the time being point to a specific isomiR. Only RNA-seq experiments would cover and quantify the entire spectrum of miR-625-3p isomiRs present. In order to answer to the question whether the miR 625-3p isomiRs that we found in the plasma are a direct consequence of the dysregulated expression pattern of cancer cells, it would be necessary to do more work, analyzing the isomiR-625-3p profile in matching tumor/non-malignant and tissue/plasma pairs by RNA-seq. Unfortunately, the TCGA-MESO cohort does not have available matched non-malignant tissue or plasma for the samples analyzed here. Therefore, this is beyond the scope of the current study and could be performed only when these sample pairs are available from the same patient. When concordance would be found, this would still be a correlation and functional investigations would be necessary.

Concerning the question whether any dysregulated miRNA expression pattern in cancer cells has been observed to be accompanied by circulating isomiRs, which can be used as biomarkers, there is at least one example. Indeed, a recent study in colorectal cancer (Roberts et al, Clin Cancer Res. 2018 24:2092-2099) has used RNA-seq analysis of plasma samples to demonstrate that levels of a miR-335-5p isomiR detects adenomas in patients under 50.

Reviewer 2 Report

In the manuscript by Kresoja-Rakic et al, authors analyzed the expression of miR-625-3p and lncRNA GAS5 in patient plasma taken before and after neo-adjuvant cisplatin chemotherapy. They try to understand the correlation between the expression changes of the ncRNAs with the clinical outcome of the patients and propose than the combined pattern change in the miR/lncRNA expression could be a signature for separation between progressive disease and stable disease/partial response. More importantly, reduced levels of GAS5 in plasma seems to indicate poor survival. I have the following major criticisms to the study

1.       I don’t know whether the authors have any particular reasons to plot deldelCT and delCT instead of standard relative expression levels. It is just making the upregulation and downregulation data less understandable and complicated. It is much easy to show the data as relative copy numbers. It is just an unwanted complexity.

2.       Again, a standard way of representing data in 2A/B will be to have box plots for change in gene expression with SD/PR and PD as the x-axis values. Especially, since they state that these differences are not statistically significant.

3.       Line 126: ‘White bar represent tissue levels of the isomiRs identified in plasma samples in our study.’ Since the authors have not performed any quantitative assays for isomiRs, I assume that what they actually mean to say is something like “White bars indicates the tissue levels of the isomiRs, which we also detected in our plasma samples”

4.       Also in the method section regarding TCGA analysis of isomirs (line 302 onwards), authors should clarify that it is extraction of data from TCGA rather than using the terms like  “detection” and “sequencing data was generated”.

5.       Line 170 onwards: Is the decrease of GAS5 correlated to survival or absolute low levels. I assume this is the direction of change (increase or decrease after chemo), rather than absolute value. This is not clearly stated. The graph in Fig 1C shows that this demarcation is necessary.

6.       The use of the term isomiRs in the title of the paper is misleading as there is no real investigation or discussion on the relevance of these in the study. Only an observation is made that they are present and is supported by TCGA. 

Author Response

Reviewer #2

 We appreciate the valuable comments and suggestions provided by the Reviewer, which are addressed below.

1.       I don’t know whether the authors have any particular reasons to plot deldelCT and delCT instead of standard relative expression levels. It is just making the upregulation and downregulation data less understandable and complicated. It is much easy to show the data as relative copy numbers. It is just an unwanted complexity.

The reasons to plot deltaCt are multiple. Having provided Ct values for normalizers, deltaCt  values allow to have a range of Ct values and this is used in recent publications on circulating ncRNA (see e.g. Poel Experimental & Molecular Medicine (2018) 50, e454). In addition, keeping in mind that Ct are on log scale, a large difference in deltaCt as we observed for miR-625-3p (Figure 1a) and relative changes in both miR-625-3p and GAS5  for a given patient (Figure 1a and c) would be more difficult to be shown in a readable manner. For these reasons, we think that deltaCt representations should be maintained in Figure 1. However, in order to address the request of the reviewer we have implemented the change from deltadeltaCt to fold expression change in Figure 2a and 2b.

2.       Again, a standard way of representing data in 2A/B will be to have box plots for change in gene expression with SD/PR and PD as the x-axis values. Especially, since they state that these differences are not statistically significant.

As mentioned in the answer to point 1, we implemented the requested change in Figure 2a and 2b and have edited the legend to the Figure accordingly.

3.       Line 126: ‘White bar represent tissue levels of the isomiRs identified in plasma samples in our study.’ Since the authors have not performed any quantitative assays for isomiRs, I assume that what they actually mean to say is something like “White bars indicates the tissue levels of the isomiRs, which we also detected in our plasma samples”

We thank the reviewer for suggesting a more understandable description. The sentence line 128 has been corrected as suggested.

4.       Also in the method section regarding TCGA analysis of isomirs (line 302 onwards), authors should clarify that it is extraction of data from TCGA rather than using the terms like  “detection” and “sequencing data was generated”.

We do apologize if the sentence could lead to misinterpretation. We have now corrected the sentence as follows (line 324):

“All small RNA sequencing data that had been generated using the Illumina HiSeq2000 platform was processed according to a previously described pipeline [47,48].”

5.       Line 170 onwards: Is the decrease of GAS5 correlated to survival or absolute low levels. I assume this is the direction of change (increase or decrease after chemo), rather than absolute value. This is not clearly stated. The graph in Fig 1C shows that this demarcation is necessary.

The whole paragraph 2.3 including the title was clearly stating that it is the direction of the change (increase or decrease after chemo) of GAS5 that is correlated with survival, therefore we think that the reviewer was mostly concerned by the discussion. Therefore we have added (lines 210 and 240) that changes in levels of GAS5 matter.

6.       The use of the term isomiRs in the title of the paper is misleading as there is no real investigation or discussion on the relevance of these in the study. Only an observation is made that they are present and is supported by TCGA. 

In order to address this concern, we removed the term “isomiRs” from the title.

Round 2

Reviewer 1 Report

I am satisfied with author's comments and modifications of the manuscript.